# E3 ubiquitin ligase Deltex facilitates the expansion of Wingless gradient and antagonizes Wingless signaling through a conserved mechanism of transcriptional effector Armadillo/β-catenin degradation

Vartika Sharma[1,2], Nalani Sachan[3], Bappi Sarkar[1], Mousumi Mutsuddi[1], Ashim Mukherjee[1]*

[1]Department of Molecular and Human Genetics, Institute of Science, Banaras Hindu University, Varanasi, India; [2]Department of Integrative Biology and Physiology, University of California Los Angeles, Los Angeles, United States; [3]Department of Cell Biology, NYU Langone Medical Center, New York, United States

*For correspondence:
amukherjee@bhu.ac.in

**Abstract** The Wnt/Wg pathway controls myriads of biological phenomena throughout the development and adult life of all organisms across the phyla. Thus, an aberrant Wnt signaling is associated with a wide range of pathologies in humans. Tight regulation of Wnt/Wg signaling is required to maintain proper cellular homeostasis. Here, we report a novel role of E3 ubiquitin ligase Deltex in Wg signaling regulation. *Drosophila dx* genetically interacts with *wg* and its pathway components. Furthermore, Dx LOF results in a reduced spreading of Wg while its over-expression expands the diffusion gradient of the morphogen. We attribute this change in Wg gradient to the endocytosis of Wg through Dx which directly affects the short- and long-range Wg targets. We also demonstrate the role of Dx in regulating Wg effector Armadillo where Dx down-regulates Arm through proteasomal degradation. We also showed the conservation of Dx function in the mammalian system where DTX1 is shown to bind with β-catenin and facilitates its proteolytic degradation, spotlighting a novel step that potentially modulates Wnt/Wg signaling cascade.

## eLife assessment

This is a **useful** study of the connection between the ubiquitin ligase protein deltex and the wingless signaling pathway. Two different links are inferred from genetic interactions in vivo between loss-of-function mutations and overexpression. While the genetic data are **solid**, the precise mechanism underlying either effect remains to be established.

## Introduction

Morphogens play a key role in cell-fate determination in a concentration-dependent manner, and their functions are indispensable to tissue patterning during development. These molecules elicit long-range signaling by forming a concentration gradient and cells differentiate based on these positional cues. The roles of morphogens such as Wingless (Wg), Hedgehog (Hh), and Decapentaplegic (Dpp) during the development of *Drosophila* appendages have been extensively studied (*Cadigan and Nusse, 1997*; *Lecuit et al., 1996*; *Nellen et al., 1996*; *Tabata, 2001*).

Wg signaling is an evolutionarily conserved pathway and has multiple roles in the regulation of a variety of developmental processes, including cell-fate specification, cell proliferation, cell survival, and migration (*Cadigan and Waterman, 2012*; *Logan and Nusse, 2004*; *MacDonald et al., 2009*; *Valenta et al., 2012*). Any aberration in the expression of candidates in the Wg signaling cascade often leads to human diseases including cancer (*Clevers, 2006*; *Coombs et al., 2008*).

In the developing wing disc, Wg is secreted from the dorsal–ventral (D/V) boundary, and a gradient of Wg is formed to elicit long-range Wg signaling (*Baena-Lopez et al., 2012*; *Diaz-Benjumea and Cohen, 1995*). Secreted Wg, consecutively induces the expression of its target Senseless (Sens) and Distal-less (Dll). High threshold Wg target gene *senseless* is responsible for the formation of margin sensory bristles of adult wings (*Couso et al., 1994*), whereas the low threshold Wg target gene *distalless* is required for proper wing growth and is expressed in a graded manner, declining toward the edges of the wing pouch (*Couso et al., 1994*; *Neumann and Cohen, 1997*). Wg also induces the expression of Cut by activating Notch in the D/V boundary (*de Celis and Bray, 1997*).

Wg exercises its effect by regulating the transcription of the target genes in the responding cells. Armadillo (vertebrate homolog β-catenin) plays a pivotal role in the transduction of Wg signaling (*Dierick and Bejsovec, 1998*; *Städeli et al., 2006*). In the absence of the Wg ligand, a protein complex comprising of APC (Adenomatous polyposis coli), Axin, GSK3-β (Shaggy), and Casein Kinase-I (Cki) mediates Armadillo (Arm) phosphorylation, and ubiquitination, followed by its degradation via the proteasome pathway. On the contrary, binding of the ligand Wg to its receptor Frizzled (Fz), at the cell surface, initiates a signaling cascade that inhibits the function of the destruction complex via adaptor protein Dishevelled (Dsh), leading to the stabilization of Arm. Arm then translocates to the nucleus thereby activating the transcription of the downstream target genes (*Mosimann et al., 2009*). In canonical Wg signaling, Arm plays a dual role. The cytosolic Arm, as mentioned, is used for the transduction of the Wg signals, whereas the Arm localized at the adherens junctions together with α-catenin and cadherin homologs is required for cell adhesion (*Peifer et al., 1993*).

We have earlier reported that *deltex* (*dx*), a known Notch signaling regulator, modulates Decapentaplegic morphogen gradient formation and affects Dpp signaling in *Drosophila* (*Sharma et al., 2022*). Here, in the present study, we report that *deltex* genetically interacts with the genes that encode Wg pathway components during wing development. Immuno-cytochemical analysis revealed that Dx facilitates the spreading of the morphogen, thereby directly affecting the expression of the short- and long-range target genes, Senseless and Vestigial. This change in the Wg gradient is attributed to the role of Dx in Wg trafficking. Here, we speculate that Dx, like in the case of Dpp and Notch, aids in the vesicular trafficking of Wg to facilitate its gradient formation. Furthermore, Dx was also seen to reduce the expression of the Wg effector Arm through proteasomal degradation. We also investigate the conservation of Dx function in β-catenin stability and degradation in cultured human cells. Here, we show that human DTX1 binds with β-catenin and facilitates its degradation. This study thus presents a whole new avenue of Dx function in the regulation of Wg signaling.

## Results

### *dx* genetically interacts with Wg pathway components

Wg controls the growth and patterning of the *Drosophila* wing and aberrant Wg signaling interferes with normal wing development. In order to ascertain the role of *dx* in Wg signaling, first, we tried to check if *dx* genetically interacts with *wg* alleles. It is interesting to note that both the wg loss-of-function alleles tested ($wg^{CX3}$ and $wg^{CX4}$) showed lethality in homozygous condition; however, in heterozygous condition, they do not exhibit any phenotype (*Figure 1B1, C1*). The two mutant alleles of *dx* (a null allele $dx^{152}$ and a loss-of-function allele *dx*) show wing vein thickening phenotype in hemizygous condition, however, $dx^{152}$ and *dx* heterozygote female flies exhibit normal wings. When the loss-of-function allele of *wg* ($wg^{CX3}$ and $wg^{CX4}$) was brought in heterozygous condition with either $dx^{152}$ or *dx* mutant alleles, they showed wings that were indistinguishable from the wild type (data not shown). However, when the dosage of *wg* was reduced in $dx^{152}$ and *dx* mutant alleles in the hemizygous background, an increased wing vein thickening phenotype was observed which is indicative of an enhanced phenotype relative to controls (*n* = 100). Moreover, an appreciable number of flies show notching in the wing margin (*Figure 1B1–C3*), suggesting that a complete absence of *dx* creates a sensitized background, which is more responsive to a lower dosage of *wg* during wing development.

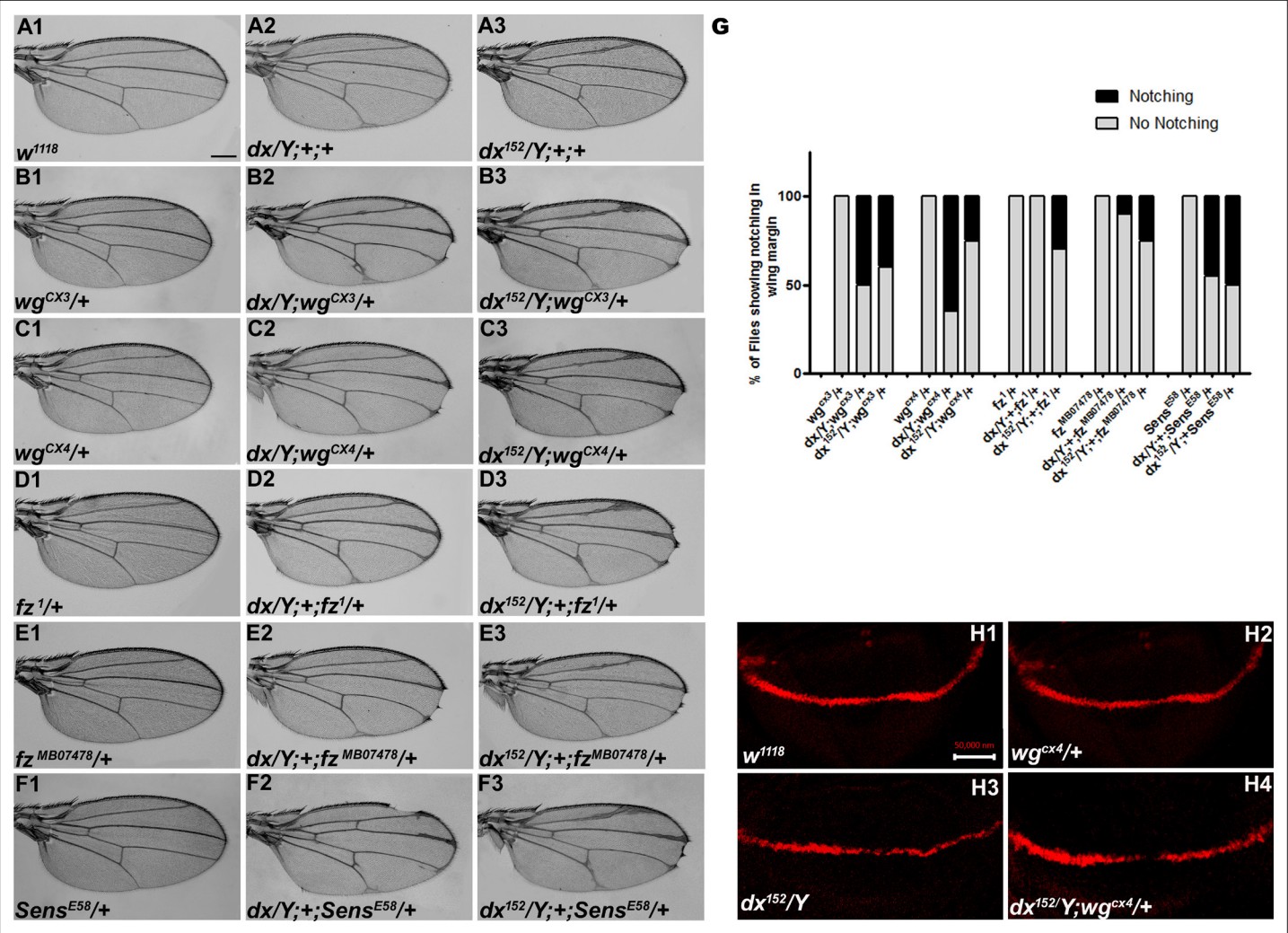

**Figure 1.** *dx* genetically interacts with Wg pathway components. (**A1–F3**) Representative wings from males with the indicated genotypes. *dx* (**A2**) and *dx^152^* (**A3**) hemizygote show extra vein material at the distal end of the wing compared to wild-type flies (*w^1118^*) (**A1**). (**B2, B3, C2, C3**) Both the *dx* allele show an enhancement of wing vein thickening along with wing notching in hemizygous combination with different alleles of wg (*wg^CX3^* and *wg^CX4^*) heterozygotes. (**D2, D3, E2, E3**) Different alleles of *fz* (*fz^1^* and *fz^MB07478^*) in trans-heterozygous conditions show enhanced vein thickening and wing nicking phenotype with *dx* hemizygotes. (**F2, F3**) *dx* alleles show strong genetic interaction with the Wg target gene *sens*, where a loss-of-function allele of sens (*sens^E58^*) shows wing nicking phenotype in flies that are homozygous for *dx* alleles. (**G**) Graph showing the frequency of wing notching phenotypes observed in indicated genetic combinations (*n* = 100). (**H1–H4**) Representative image of Cut in wing imaginal disc in different genetic combinations. (**H2**) The expression of Cut in *wg^CX4^* heterozygote is similar to the wild-type Cut expression (**H1**). (**H3**) *dx^152^* hemizygotes show a reduction in Cut expression in the dorsal–ventral (D/V) boundary of the wing disc. (**H4**) Wing disc from *dx^152^/Y; wg^CX4^/+* genotype shows a further reduction in Cut expression when compared to *dx^152^/Y* wing discs. Images in H1–H4 are representatives of three independent experiments (*n* = 6). Scale bar: A1–F3: 200 μm. H1–H4: 50 μm.

Furthermore, the expression of Cut, a downstream target of Wg signaling activity (*Katanaev et al., 2008*) at the D/V boundary of the wing disc, was found to be reduced in wing discs from these larvae. Third instar larval wing discs displayed an interrupted Cut expression at the intersection of A/P and D/V boundary from individuals carrying heterozygous *wg* loss-of-function allele in *dx* hemizygous null background (*Figure 1H1–H4*). These results clearly indicate that *dx* may play a role in *Drosophila* Wg signaling activity.

Furthermore, we tried to elucidate if the receptor of *wg*, *frizzled* (*fz*), does exhibit any interaction with *dx*. When the hypomorphic allele of *fz* (*fz^1^*), was brought in heterozygous condition with *dx^152^* null, the male flies displayed wing notching phenotype in addition to wing vein thickening (*n* = 100) (*Figure 1D1–E3*). *senseless* (*sens*), a short-range target of Wg signaling also displayed a strong genetic interaction with *dx*. A loss-of-function allele of *sens* (*sens^E58^*) is recessive lethal in homozygous

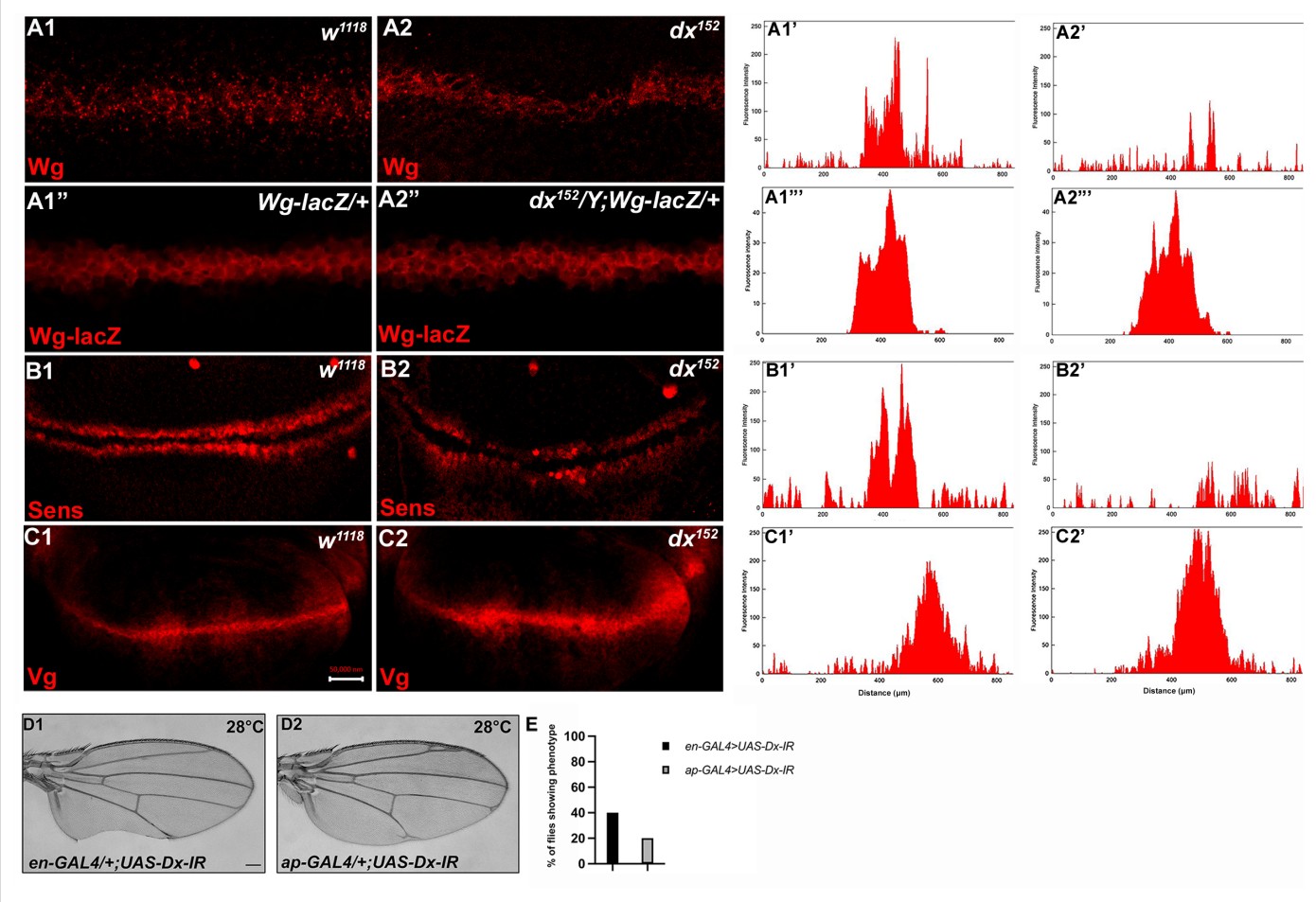

**Figure 2.** Loss of Dx reduces Wg signaling gradient and target gene expression. (**A2**) *dx^152* wing disc shows a narrow Wg expression gradient compared to wild-type third instar larval wing discs (**A1**). (**A1″, A2″**) show Wg-lacZ staining in the mentioned genotype. (**B2**) A constricted expression of Sens was observed in dx null discs (*dx^152*) (marked by arrowhead) compared to the control wild-type wing imaginal discs (**B1**). (**C1**) The expression of Vg in the wild-type disc. (**C2**) dx null discs showed a broadened Vg expression, marked by arrowheads. (**A1′–C2′**) Show the average fluorescence intensity of images in A1–C2. (**D1, D2**) Expression of the RNAi targeting Dx by *en-GAL4* or *ap-GAL4* results in morphological aberrations in the expressing regions. (**E**) Graph showing the percentage of flies showing respective phenotypes in the mentioned genotype (*n* = 100). Images in A–C are representatives of three independent experiments (*n* = 6). Scale bar: A1–A2: 10 μm. B1–C2: 50 μm. D1, D2: 200 μm.

condition; however, in heterozygous condition, they do not display any phenotype. On the contrary, on reducing the dosage of *sens* in *dx* null and *dx* loss-of-function allele in hemizygous condition, an enhanced wing phenotype with marked serration in the wing margin was observed (*n* = 100) (*Figure 1F1–F3*). These results displayed a strong genetic interaction between *dx* and Wg pathway components, suggesting that Dx may have functional implications in *Drosophila* wing development.

## Loss of *dx* reduces Wg signaling

To better understand the role of *dx* in Wg signaling regulation, we studied the expression of Wg and its target genes in *dx* null wing discs. It was found that in *dx* null third instar larval wing discs, the gradient of Wg at the D/V boundary was shortened. High-magnification pictures revealed a significant decrease in the amount of Wg emanating from the Wg-producing cells at the D/V boundary (*Figure 2A2*). Thus, Dx is required for the proper Wg gradient formation, as the range of Wg expression and the number of Wg puncta was significantly reduced in *dx* null condition. Similar interruption in Wg gradient was also observed by Hori et al., where *dx^24* allele showed a reduction in Wg expression (*Hori et al., 2004*), however, the study focuses on the indispensable role of *dx* in Notch signaling.

We also analyzed the expression of Wg target genes in *dx* null discs. The domain of Vestigial, the long-range target of Wg was found to be significantly broadened in *dx* null discs (*Figure 2C2*). This

is in contrast to the reported partial suppression of Vg expression in *ptc-GAL4* expressed Dx dominant negative discs (*Hori et al., 2004*). Moreover, in another set of experiments, we found that the expression of the short-range target Sens showed slight deformity in the expression pattern with a constricted expression at the D/V boundary (*Figure 2B2*).

Expression of the short-range target such as Sens needs a higher threshold of Wg morphogen gradient than the long-range target like Vestigial which often requires a less concentrated Wg gradient. Thus, our results show that a proper Wg gradient is not formed in the absence of Dx, and this supports that Dx may have a profound role in regulating Wg signaling through its gradient formation.

## Dx regulates Wg signaling

Based on the loss-of-function genetic interaction studies between *dx* and *wg* mutant alleles that suggest the role of Dx in Wg signaling, we further tried to investigate the gain-of-function effect of Dx on Wg signaling activity. For this, *UAS-GAL4* binary system was exploited. We over-expressed FLAG-tagged *UAS-Dx*, with *engrailed-GAL4* (*en-GAL4*) driver strain, that drives the expression of the transgene in the posterior compartment of the wing imaginal disc, and in turn in the adult wing. Overexpression of *Dx* with *en-GAL4* shows pupal lethality at 25°C, however, at 18°C, the flies emerge with massive morphological defects. The wing displayed blisters along with vein loss (*Figure 3—figure supplements 1–A2*). Besides, Dx over-expression with *apterous-GAL4* (*ap-GAL4*) also resulted in abnormal wings that failed to appose properly and show large blisters (*Figure 3—figure supplement 1–A3*), and since Wg signaling modulates the expression of cell adhesion molecules in *Drosophila* wing imaginal disc cells (*Bienz, 2005*; *Wodarz et al., 2006*) such phenotypes are often associated with impaired Wg signaling.

To check if Dx is involved in Wg pathway regulation in other developmental contexts, we over-expressed Dx in the primordia of the dorsal thorax using *ap-GAL4* and *pnr- GAL4*. Dx over-expression in the thorax leads to reduced and disoriented macrochaetae and reduction in scutellar bristles (*Figure 3—figure supplements 1–B2–B3*). A reduced tarsal segment along the proximodistal axis and dorsoventral shift of the sex combs was observed when Dx was over-expressed with *distal-less-GAL4* (*dll-GAL4*) in the leg (*Figure 3—figure supplements 1–C2*). All these phenotypes are characteristics of reduced Wg signaling as also shown earlier for ectopic expression of another gene required for epithelial planar polarity, *nemo* (*Zeng and Verheyen, 2004*). Dx shows a similar phenotype when over-expressed in flies suggesting that like *nemo*, Dx may antagonize the Wg signaling activity.

To further test whether Dx affects the Wg signaling output we tried to analyze the expression of the Wg target genes in the wing imaginal discs of the third instar larvae. Wg activates the expression of the proneural gene *senseless* (*sens*) in the sensory organ precursor, which later develops into bristles of the adult wing margin (*Blair, 1996*; *Couso et al., 1994*). The expression of Senseless, a canonical target normally activated by high Wg levels in two discrete stripes at the D/V boundary (*Nolo et al., 2000*) was found to be significantly reduced in the posterior compartment where Dx was over-expressed (*Figure 3A2*). It is known that Wg can also indirectly induce the expression of Cut. A positive feedback loop between Wg expressing cells along the D/V boundary and Serrate and Delta expressing cells in adjacent cells maintains the Notch signaling activity along the D/V boundary, which consecutively induces the expression of Cut (*de Celis and Bray, 1997*). Similar to Sens, we found that the expression of Cut was also drastically abolished upon Dx over-expression (*Figure 3B2*). Moreover, to the contrary, the expression of Vestigial (Vg), which is activated in response to low ligand concentrations (*Neumann and Cohen, 1997*; *Zecca et al., 1996*) was found to be expanded in the posterior domain of the wing imaginal disc (*Figure 3C2*), indicating that Dx reduces the Wg gradient. These results strongly suggest that Dx modulates Wg signaling output.

Also, in our earlier reports, we demonstrated the role of Dx in inflicting apoptosis. Dx synergizes with JNK (c-Jun N-terminal Kinase) pathway ligand Eiger and triggers caspase-dependent cell death (*Dutta et al., 2018*). Similar synergistic action of Dx was also reported with TRAF6 where the association triggers JNK activation in an Eiger-independent manner (*Sharma et al., 2021a*). Thus, we tried to dissect if the loss of these markers (Sens and Cut) is due to cell death. We inhibited apoptosis with caspase inhibitor protein p35 in Dx over-expression background and no rescue in the level of Sens and Cut expression was observed in the posterior compartment of the wing disc (*Figure 3D1–E4*) suggesting the loss of Sens and Cut is independent of cell death.

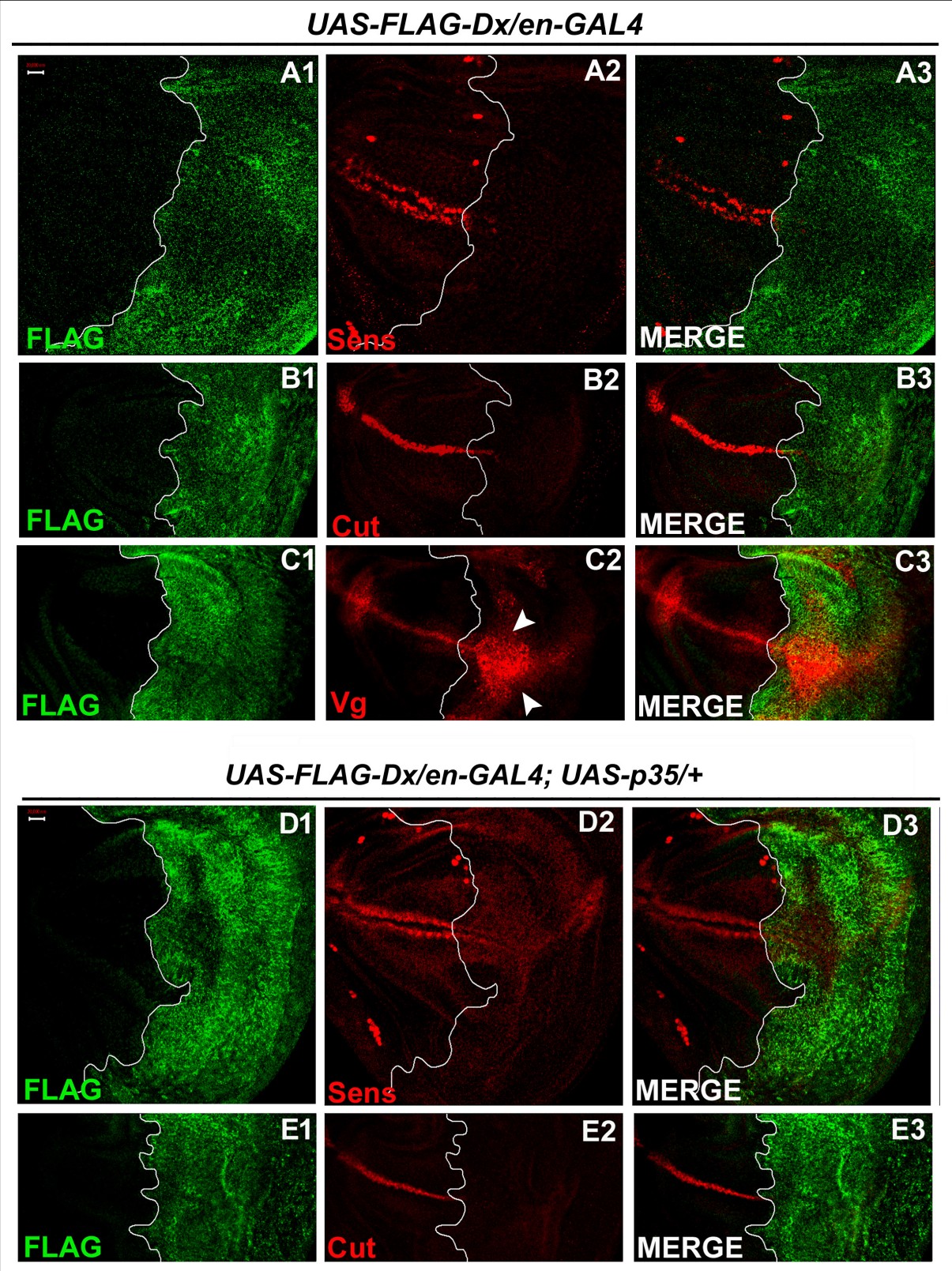

**Figure 3.** With two supplements: Dx modulates the expression of the Wg target genes. (**A1–A3**) Over-expression of Dx using posterior domain-specific *en-GAL4* results in a complete loss of Senseless (**A2**). Third instar larval wing discs of the indicated genotypes were immunostained to detect FLAG (green) and Senseless (red). (**B1–B3**) A reduction in expression of Cut was observed in the posterior domain of the wing disc upon Dx over-expression with *en-GAL4*. (**C1–C3**) Expression of Vestigial (Vg), on the contrary, was found to be expanded in the posterior compartment. Arrowheads indicate the

*Figure 3 continued on next page*

*Figure 3 continued*

expansion of Vg expression close to the anterior–posterior (A/P) boundary of the wing imaginal disc (**C2**). (**D1–E3**) Over-expression of caspase inhibitor p35 together with Dx does not show any rescue in the expression of Senseless (**D2**) and Cut (**E2**) suggesting the event is independent of apoptosis. The white line marks the boundary of FLAG expression. Images in A–E are representatives of three independent experiments (*n* = 6). Scale bar: A1–E3: 50 μm.

The online version of this article includes the following figure supplement(s) for figure 3:

**Figure supplement 1.** Ectopic Dx mimics Wg loss-of-function phenotype.

**Figure supplement 2.** Dx-mediated Wg regulation is independent of Notch.

It is worth mentioning here that similar to our findings, the expression of Sens was found to be down-regulated when Reggie-1, a component of the membrane micro-domain was up-regulated. The reduction in Sens and Cut expression, in their studies, was attributed to an enhanced spreading of the morphogen, Wg (*Katanaev et al., 2008*).

*Drosophila* Dx interacts with the ankyrin-rich repeats of the Notch intracellular domain and regulates Notch signaling positively (*Matsuno et al., 1995*). Moreover, negative regulation of Dx in association with non-visual β-arrestin, Kurtz was reported earlier, where polyubiquitination-mediated degradation of Notch by Dx was attributed to this loss (*Mukherjee et al., 2005*). A similar reduction of Notch and its targets was reported lately when Dx synergistically interacts with TRAF6 and Hrp48 (*Dutta et al., 2017*; *Mishra et al., 2014*). Human Deltex (DTX1) was also shown to regulate NOTCH1/HES1 signaling negatively in osteosarcoma cells (*Zhang et al., 2010*). It was reported earlier that in D/V boundary of the wing disc, the regulation of Wg is Notch dependent (*Neumann and Cohen, 1996*), we tried to elucidate if the regulation of Wg via Dx is through Notch. Thus, we examined the status of Deadpan (Dpn) and NRE (Notch response element), a direct target of Notch (*Zacharioudaki and Bray, 2014*). In contrast to reduced levels of Sens and Cut, the expression of Dpn was found to remain unaltered in Dx over-expressed conditions (*Figure 3—figure supplement 2A1–A4*), suggesting that Dx can possibly regulate the two signaling cascades, viz. Notch and Wg independent of each other. Furthermore, we also checked the status of the Notch response element (NRE-GFP) in Dx over-expression background and found a punctate NRE expression in the posterior domain (*Figure 3—figure supplement 2B1–B4*). The role of Dx in Notch regulation is evident however if Dx regulates Wg through Notch or if the two events are independent is still a crucial aspect to investigate. Our results however indicate an independent role of Dx in Wg signaling regulation. We however do not rule out the alternate possibility of Wg regulation through Dx via Notch.

## Over-expression of Dx expands the Wg diffusion gradient

In the third instar wild-type wing discs, Wg is secreted from the producing cells at the D/V boundary, which diffuses on either side to form a concentration gradient. Over-expression of Dx, by the A/P boundary-specific *ptc-GAL4*, results in an erosion of the Wg gradient (*Figure 3A2 and B2*). Wg expression can be visualized as spreading far into the disc from the site of production. A high-magnification picture further revealed far-spreading punctate Wg expression more significantly in the ventral compartment. It is interesting to note that the number of Wg puncta and their apparent size is also significantly increased when Dx was over-expressed using *ptc-GAL4* (*Figure 4B2*, *Hori et al., 2004*). Similar diffusion of Wg gradient was also observed when Dx was over-expressed with posterior domain-specific *en-GAL4* driver (*Figure 4C2*).

We also observed a significant increase in the extracellular Wg expression upon Dx over-expression further corroborating our initial results (*Figure 4D2*). Earlier it was shown that Swim (Secreted Wg interacting molecule) promotes long-range Wg signaling by maintaining Wg solubility (*Mulligan et al., 2012*). The report postulates that Swim reduction results in an increase in extracellular Wg which consequently increases the expression of long range Wg target, Dll (*Mulligan et al., 2012*). We have observed a very similar phenotype with another long range Wg target, Vg in increased Dx expression condition (*Figure 3C2*). Here, we do not rule out the possibility that Dx potentially aids in Wg mobilization from the adherens junction. Several models from studies in *Drosophila* embryonic epidermis and wing disc have been put forward to explain Wg movement across the cells (*Cadigan, 2002*; *Tabata and Takei, 2004*). Our result indicates that Dx facilitates Wg spreading which flattens its gradient. This, in turn, results in the reduction of the short-range Wg targets (*Figure 3A1–A3*), which

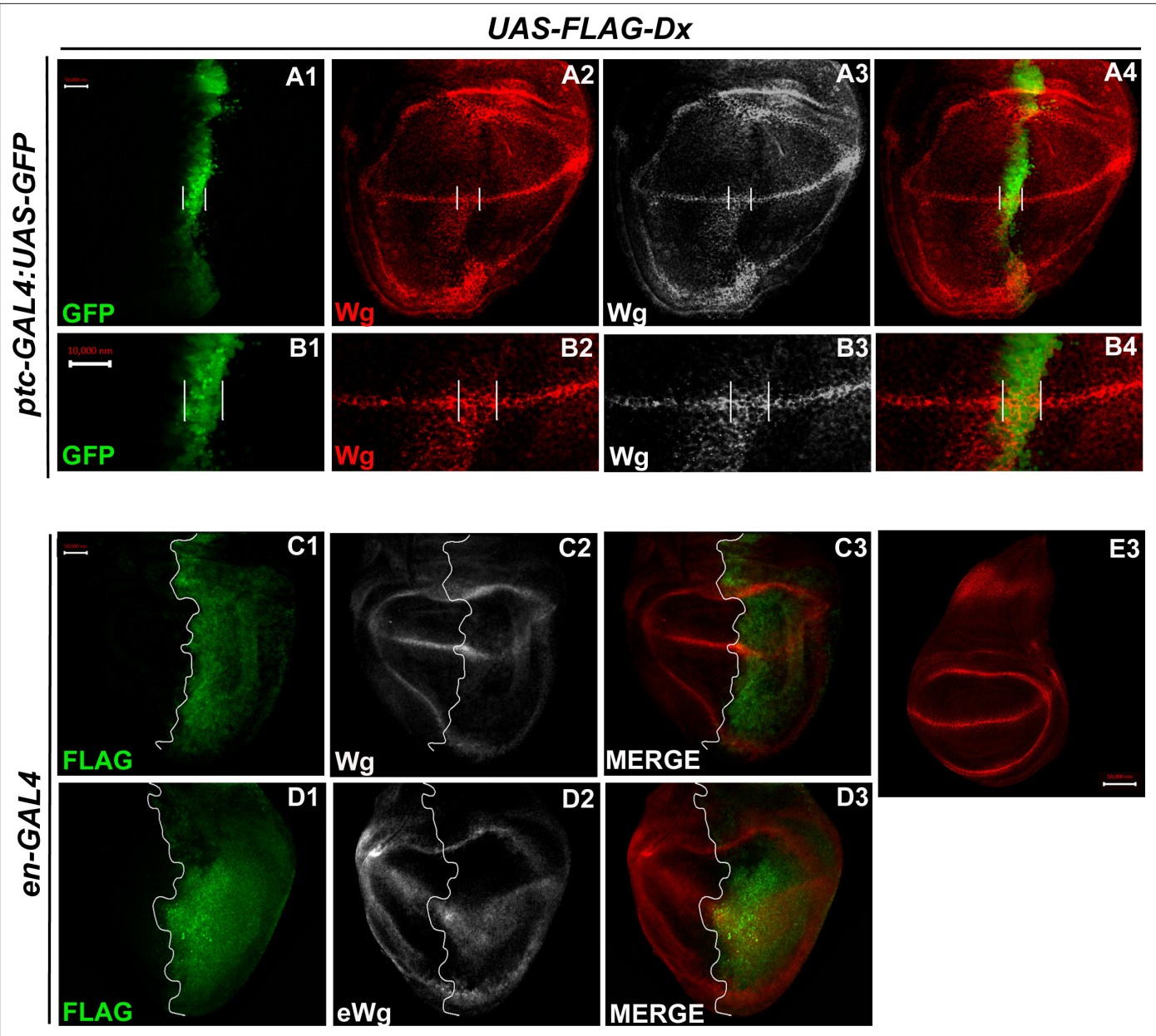

**Figure 4.** Over-expression of Dx expands the Wg diffusion gradient. (**A1–B4**) Over-expression of Dx by anterior–posterior (A/P) domain-specific *ptc-GAL4* results in the broadening of the Wg diffusion domain (**A2**). Grayscale images are for better contrast (**A3**). (**B2**) High-magnification picture shows increased Wg puncta at the A/P–dorsal–ventral (D/V) junction with more significant expression in the ventral portion of the disc (marked by arrowhead) (**B2, B3**). The white line marks the Ptc domain, respectively. (**C1–C3**) Dx over-expression with *en-GAL4* results in a diffused Wg expression in the posterior domain of the wing disc. (**D1–D3**) A marked increase in extracellular Wg (eWg) was observed upon Dx over-expression with *en-GAL4*. (**E3**) Representative image of wild-type Wg expression. Images in A–D are representatives of three independent experiments (*n* = 6). Scale bar: A1–A4, C1–D3, and E: 50 µm. B1–B4: 10 µm.

often require high levels of Wg. On the contrary, the long-range Wg target showed an expansion in their expression domain (*Figure 3C1–C3*), suggesting that Dx affects the Wg gradient formation.

## Dx-mediated Wg signaling is regulated by endocytosis

Extensive evidence suggests that signaling and endocytosis are intricately linked. Classically, endocytosis has been thought to reduce the number of accessible receptors, either through its sequestration in the endosomal vesicles or degradation in lysosomes, which consecutively down-regulates the signaling. However, some reports support the notion that endocytosis can also positively regulate

the signaling cascade, its initiation, and propagation (*Hupalowska and Miaczynska, 2012*; *Seto and Bellen, 2006*).

Different mechanisms have been suggested for the role of endocytosis in various signaling cascades. Notch is continuously internalized into early endosomes and followed by its internalization, Notch is sorted to other endocytic compartments including multivesicular bodies/late endosomes, and lysosomes, or recycling endosomes (*Fortini and Bilder, 2009*; *Sachan et al., 2023*). Thus, this step helps to maintain a quantitative as well as a qualitative pool of Notch. Once endocytosed, Notch follows a vesicular trafficking pathway that determines its final fate. In the canonical pathway, Notch gets internalized after binding to its ligand. However, during the non-canonical pathway, Notch can be internalized in the absence of ligand interaction. Once internalized, the Notch protein faces almost the same pathway that begins with its incorporation into the early endosomal vesicles. From here, it is either recycled back to the cell membrane, or it becomes a part of late endosomal vesicles/mature endosomes (*Fortini and Bilder, 2009*; *Vaccari et al., 2008*; *Wilkin et al., 2008*; *Yamada et al., 2011*).

In Notch trafficking and endocytosis, Dx plays a key role. Dx facilitates the trafficking of Notch receptors from the cell membrane into the early and then to the late endosomal vesicles in the cytoplasm (*Hori et al., 2004*). It was observed that Dx, directly or indirectly, increased the half-life of these accumulated Notch proteins in the endosomal vesicles (*Hori et al., 2004*). If the transport of Notch into the late endosome was blocked, it hampered Dx-mediated Notch signal activation (*Hori et al., 2004*). Further study, using a similar approach, showed that Dx genetically interacts with the components of HOPS and AP-3 endocytic complexes that, successively, have a significant influence on Notch signaling (*Wilkin et al., 2008*). Dx has also been shown to play a critical role in Dpp endocytosis and trafficking (*Sharma et al., 2022*).

Like Notch and Dpp, evidence supporting Wg endocytosis has been reported earlier (*Seto and Bellen, 2006*). Wg is found to co-localize with Rab5 and Rab7 in the early and late endosomes, respectively (*Marois et al., 2006*). We also observed a significant co-localization of Wg with Dx in Rab5-positive vesicles (*Figure 5A1–A4*). Rab5 is a small GTPase that is required for endosome fusion to form early endosomes (*Clague and Urbé, 2001*). Impaired endocytosis affects Wg trafficking and in turn the signaling. Expression of a Dominant negative form of Rab5 (Rab5$^{S43N}$) blocks the early steps of endocytosis, including internalization and early-endosome fusion (*Entchev et al., 2000*), and significantly affects the Wg signaling (*Seto and Bellen, 2006*). A more punctate Wg expression was observed when the endocytosis is blocked by over-expressing a GDP-bound Rab5 (Rab5DN or Rab5$^{S43N}$) with a D/V boundary-specific *C96-GAL4* (*Figure 5C1*). A further reduction in Wg puncta was observed when Dx was over-expressed in the same background, suggesting the role of Dx in Wg trafficking (*Figure 5C2*).

Additionally, *C96-GAL4*-mediated over-expression of Rab5DN, results in a loss of wing tissue similar to the loss of Wg, implying the role of endocytosis in Wg signaling (*Baker, 1988*; *Couso et al., 1994*). This loss of wing tissue was further aggravated when Dx was over-expressed along with Rab5DN suggesting a more diffused Wg expression at the D/V boundary, which further limits the threshold required to form the adult wing margin (*Figure 5B2*). This synergism brings about 90% lethality, further implicating the role of Dx in Wg regulation via endocytosis. In addition to Rab5, the late endosome protein Rab7, when over-expressed with Dx, showed a punctate Wg expression (*Figure 5F2*), suggesting an important role of Dx in Wg trafficking. Rab7 targets the endocytic cargo from the early endosome to the late endosome and then to the lysosome for degradation. Over-expression of a wild-type Rab7 increases trafficking toward the lysosome. This is due to enhanced vesicle fusion, resulting in enlarged late endosomal structures, prone to lysosomal degradation (*Entchev et al., 2000*). When UAS-Rab7 was co-expressed with Dx, a more punctate Wg expression was observed evincing that Dx might facilitate the functions of Rab7. Our study suggests the involvement of Dx in Wg endocytosis, however, further analysis is required to unravel the detailed mechanistic aspect in Wg endocytosis and trafficking via Dx.

## Dx modulates Wg signal transduction

Armadillo, the *Drosophila* ortholog of mammalian β-catenin has a dual function. It acts as a nuclear transcription factor that transduces the Wg signaling on one hand and maintains the cell integrity on the other (*Coombs et al., 2008*). Within the wing disc, Arm concentrates apically where it binds with the transmembrane cadherins to build up the adherens junction that connects the actin filaments

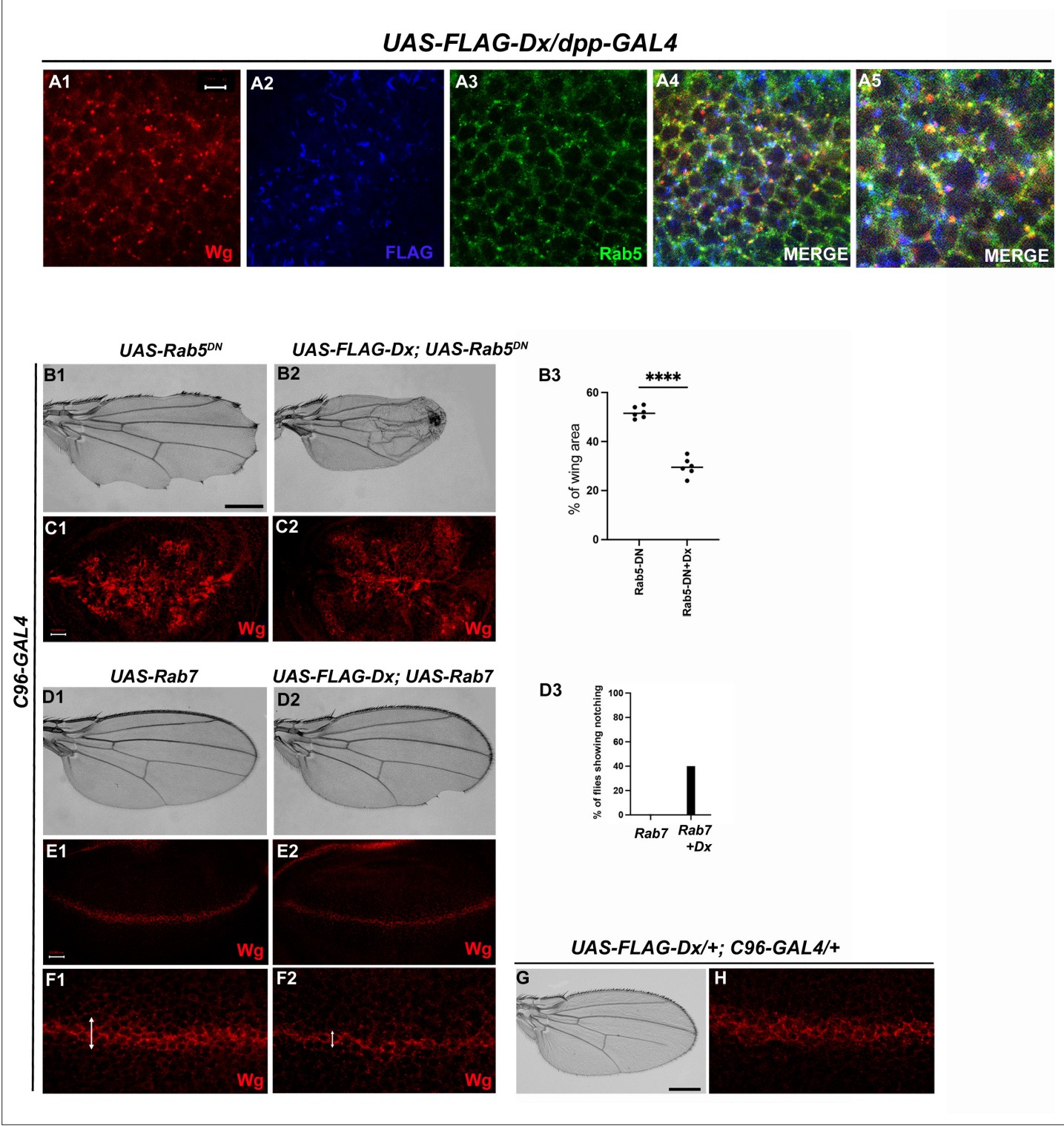

**Figure 5.** Dx-mediated Wg signaling is regulated by endocytosis. (**A1–A5**) Dx (blue) co-localizes with Rab5 (green) and Wg (red) in the same subcellular compartment. The arrows mark the co-localized spots. (**A5**) High-magnification image of (**A4**). (**C1**) Over-expression of a Dominant Negative Rab5 with *C96-Gal4* inhibits the endosomal fusion at the dorsal–ventral (D/V) boundary, enhancing the Wg diffusion gradient. (**C2**) Co-expression of Dx with Rab5DN shows a further enhancement of the Rab5DN phenotype in the Wg diffusion gradient. (**B1, B2**) Representative adult wing image of the mentioned genotype (*n* = 50). (**B1**) Over-expression of Rab5DN results in wing tissue loss consistent with decreased Wg signaling. (**B2**) A further reduction in wing size is observed upon co-expressing Dx with Rab5DN. (**B3**) The graph shows the wing area percentage of the mentioned genotypes (*n* = 6). ****p < 0.0001; unpaired *t*-test. (**E1, F1**) Over-expression of Rab7 with *C96-GAL4* renders no significant change in Wg expression. (**E2, F2**) Rab7

*Figure 5 continued on next page*

*Figure 5 continued*

when co-expressed with Dx enhances the Wg spreading. (**D1, D2**) Representative adult wing of defined genotype (*n* = 50). Note the notching phenotype in the wing upon over-expression of FLAG-Dx together with UAS-Rab7 (**D2**). (**D3**) The graph shows the percentage of flies showing notching phenotype (*n* = 100). (**G, H**) Representative adult wing image of *UAS-FLAG-Dx* over-expressed with *C96-GAL4*. (**H**) Wg expression in *UAS-FLAG-Dx>C96 GAL4* discs. Images in C and E are representatives of three independent experiments (*n* = 6). Scale bar: A1–A4: 5 µm. C1, C2 and E1, E2: 50 µm. F1, F2: 20 µm. B1, B2 and D1, D2: 200 µm.

across polarized epithelial cells, thereby acting as a bridge between E-cadherin and the actin cytoskeleton. To investigate the functional aspect of Dx in Wg signaling regulation, we analyzed the status of this key effector of Wg signaling. Upon Dx over-expression with *en-GAL4*, the accumulation of Arm was strongly reduced in the whole posterior compartment (*Figure 6A2*), and most evidently at the D/V boundary. A similar reduction in Arm expression was observed in the eye disc in the GMR domain (*Figure 6—figure supplement 1B1*) indicating that Dx might disrupt the cell-fate specification in the eye tissue. Moreover, confocal Z stack analysis revealed a reduction of apical Arm accumulation (*Figure 6B2*). The reduction of the apical Arm is consistent with the decline of cell adhesion, leading to the structural reorganization of the cytoskeleton (*Wodarz et al., 2006*). Higher magnification pictures further revealed a disorganized cell morphology upon Dx over-expression. Moreover, it was also observed that Dx co-localizes with endogenous Arm (marked by arrow) and is in the same vesicle upon Dx over-expression, suggesting that the two proteins might physically interact (*Figure 6C3, C4*).

Furthermore, upon over-expressing activated Arm (Arm$^{s10}$), with the wing margin-specific *C96-GAL4*, ectopic hairs were observed throughout the wings, and this is more evident near the wing margin (*Figure 6E1, E2*). This is reported ro be a Wg gain-of-function phenotype (*Vuong et al., 2018*). The ectopic hair phenotype, however, was significantly reduced when Dx was over-expressed in the same background (*Figure 6E5, E6*), corroborating with our immuno-staining results. A similar reduction in Arm gain-of-function phenotype was observed when the activated Arm was over-expressed together with Dx in the *Drosophila* eye. Over-expressing Arm$^{s10}$ with eye-specific *GMR-GAL4* renders flies with reduced eye size (*Freeman and Bienz, 2001*) and severely diffused ommatidia (*Figure 6F1, F2*). Moreover, upon over-expressing Dx in the same background, the Arm gain-of-function phenotype was found to be suppressed (*Figure 6F5, F6*).

Interestingly, it was also observed that Dx co-localizes with endogenous Arm and is in the same vesicle upon Dx over-expression (*Figure 6C3*). Since Arm accumulation was reduced in the cytoplasm of Dx over-expressed cells in the disc, we also checked if there is any alteration of nuclear localization of Arm in these cells. To investigate this, we analyzed the nuclear versus cytoplasmic localization of Arm protein in the wing imaginal disc. High-magnification pictures gave us a hint of Arm being exclusively in cytoplasm in Dx over-expressed condition. This inference was made since no localization of Arm in the nucleus (DAPI-stained area) was observed in the posterior compartment of the wing disc, though the anterior compartment showed significant localization of Arm with DAPI (marked by arrow) (*Figure 6—figure supplement 2A1–A10*). The small size of cells in the wing disc made it difficult to draw a definite result and a more robust and facilitated assay system is required to visualize the changes i.e. cytoplasm versus nuclear localization.

Since Dx possesses an E3 ubiquitin ligase activity, so this down-regulation can be a consequence of ubiquitination-mediated degradation of Arm which needs to be determined.

## Dx degrades Armadillo by the proteasome-mediated mechanism

The stabilization of Arm/β-catenin is critical in Wg/Wnt signaling regulation and the ubiquitination-dependent proteasomal degradation is considered to be a major mechanism for controlling its stability. β-TrCP, Jade1, casitas B-lineage lymphoma (c-Cbl), and SIAH1 [the human homolog of *Drosophila* seven in absentia (sina)], are known E3 ligases that mediate β-catenin ubiquitination in the cytoplasm (*Spiegelman et al., 2000*). Moreover, β-catenin ubiquitination in the nucleus is mediated by c-Cbl and TRIM33 (tripartite-motif containing protein 33). C-Cbl is a RING finger E3 ubiquitin ligase that binds to the non-phosphorylated nuclear β-catenin via the armadillo repeat region (*Chitalia et al., 2013*). Though the ubiquitination of β-catenin is not always linked with its degradation and there are reports where E3 ubiquitin ligases enhanced the stability of β-catenin by ubiquitinating through Lys-29- or Lys-11-linked ubiquitin chains (*Hay-Koren et al., 2011*). Thus, to elucidate if Arm down-regulation through Dx is a ubiquitination-dependent phenomenon, first, we tried to test if Dx physically interacts

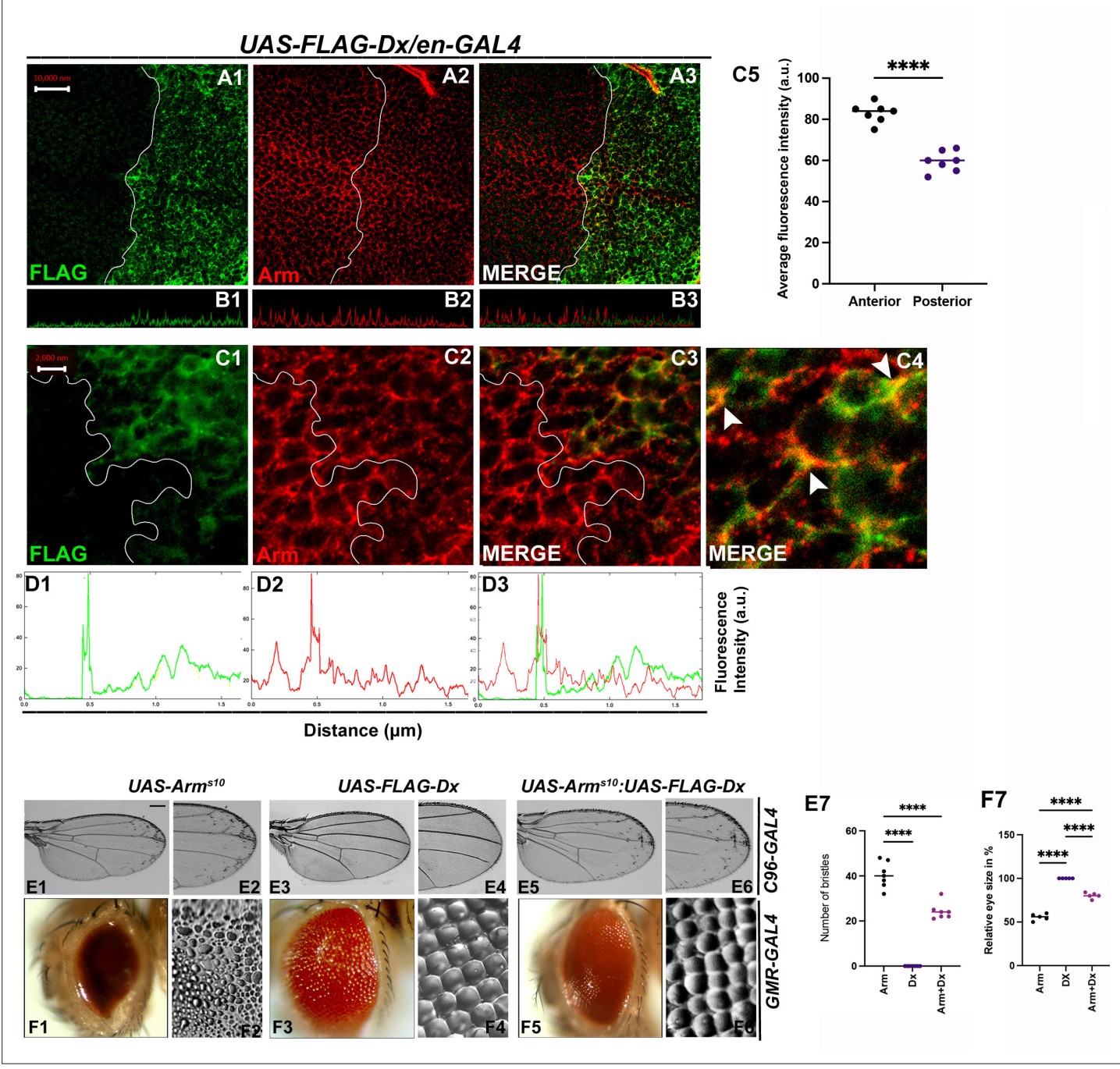

**Figure 6.** With 2 supplements: Dx modulates Wg Signal transduction. (**A1–C4**) Dx over-expression with *en-GAL4* in the wing imaginal disc results in a reduction of endogenous Armadillo throughout the disc with a more pronounced reduction at the dorsal–ventral (D/V) boundary. (**B1–B3**) Confocal Z stack Intensity profiling shows a reduced Arm level in the posterior compartment of the disc. (**C1–C4**) A higher magnification picture shows a significant overlap between FLAG-tagged Dx and endogenous Arm (marked by arrowhead). Note the reduction of Arm in the posterior compartment of the disc. (**C5**) Graph shows the average Arm fluorescence intensity (a.u., arbitrary units) ($n = 7$). ****$p < 0.0001$; unpaired *t*-test. (**D1–D3**) Show the average FLAG and Arm fluorescence intensity profiles for the representative wing disc. (**E1–E6**) Representative adult wing of defined genotype ($n = 50$). (**E1, E2**) Over-expression of the activated form of Arm (Arm$^{S10}$) with *C96-GAL4* shows ectopic hairs more significantly along the wing margin of the adult wing. (**E3, E4**) Dx over-expression in the wing shows no ectopic hairs. (**E5, E6**) Co-expression of Dx with ArmS10 results in the suppression of Arm gain-of-function effect and a relative reduction in the number of ectopic bristles was observed. (**E7**) Graph shows the number of bristles in each combination. ****$p < 0.0001$; one-way analysis of variance (ANOVA)/Turkey's multiple comparisons test. (**F1–F6**) Representative adult eye and eye imprint of defined genotype ($n = 10$). (**F1, F2**) GMR-specific expression of an activated form of Arm produces a severely reduced eye with diffused ommatidia. (**F3, F4**) Dx over-expression renders no significant phenotype. (**F5, F6**) Arm over-expression phenotype gets dramatically suppressed when Dx was co-expressed in the

*Figure 6 continued on next page*

*Figure 6 continued*

same background. (**F7**) Graph represents the phenotypic score of the eye size (*n* = 6). ****p ,0.0001; one-way analysis of variance (ANOVA)/Turkey's multiple comparisontest. Images in A–C are representatives of three independent experiments (*n* = 6). Scale bar: A1–A3: 10 µm. C1–C3: 2 µm. E1–E6 and F1–F6: 200 µm.

The online version of this article includes the following figure supplement(s) for figure 6:

**Figure supplement 1.** Dx modulates Arm expression in the eye tissue.

**Figure supplement 2.** Dx might facilitate the cytoplasmic retention of Arm.

with Arm. Co-immunoprecipitation experiments using FLAG-tagged Dx revealed that FLAG-tagged Dx pulled down Arm when co-expressed in eye tissue (*Figure 7A*). Likewise, FLAG-tagged Dx was coimmunoprecipitated with Arm using an anti-Arm antibody when the two proteins were expressed together (*Figure 7B*).

Furthermore, to investigate the role of Dx in Arm degradation we incubated the imaginal discs with MG312, a proteasome inhibitor, before performing Arm staining. A rescue in Arm loss post-MG132 treatment was observed in the posterior domain of the wing disc suggesting it to be a ubiquitination-dependent phenomenon (*Figure 7C1–E4*). Moreover, Western blot reconfirmed our immunofluorescence studies where a decline in Arm expression was observed in Dx over-expression which was significantly rescued after MG132 treatment (*Figure 7G, H*). These results unravel a novel mechanistic aspect of Dx in Wg signaling regulation.

## Conserved mechanism of β-catenin degradation by human DTX1

In order to elucidate the conserved role of Dx in Wg signaling regulation across the phyla, we examined the status of β-catenin in human DTX1over-expressed HEK-293 cells. In the *Drosophila* system, we have observed that Arm physically interacts with Dx and proteasome inhibitor, MG132, treatment resulted in stabilization of Arm in Dx-over-expressed wing imaginal discs. Thus, to determine the functional conservation of human homolog of *Drosophila* Dx, DTX1 in the β-catenin degradation, first we checked the binding of human DTX1 with β-catenin in HEK-293 cells through co-immunoprecipitation experiment. Consistent with the *Drosophila* studies, the results here undoubtedly reveal the fact that DTX1 binds to β-catenin (*Figure 8A*). In addition, as in the case of *Drosophila*, β-catenin also gets stabilized in the presence of MG132 in human HEK-293 cells over-expressing DTX1. Post-MG132 treatment, the HEK-293 cell lines transfected with HA-tagged DTX1 showed a rescue in β-catenin expression (*Figure 8B*). These results clearly indicate that there is a conserved mechanism through which DTX1 degrades β-catenin and in turn regulates Wnt signaling.

## Discussion

The diverse role of Wg in different developmental contexts depends on the Wg signaling activity. Endocytosis and intracellular trafficking regulate Wg signaling levels. Internalization and endosomal transport of Wg not only affect levels and distribution of Wg ligand but also regulates signal transduction through Arm stabilization. Regulation of Wg pathway is tightly monitored at multiple steps. Canonically, Wg exerts its effects by regulating the transcription of the target genes through the effector molecule, Arm (*Dierick and Bejsovec, 1998*; *Städeli et al., 2006*). In the absence of the Wg signaling the multi-protein destruction complex (APC, Axin, GSK3-β, and Cki) phosphorylates and degrades Arm. However, in the presence of the ligand Wg, the cell surface primes a signaling cascade with the aid of the scaffold protein, Axin, which in turn inhibits the destruction complex through the adaptor molecule Dishevelled leading to the stabilization of Arm. The stabilized Arm thereby enters the nucleus to activate the transcription of the target genes (*Bilic et al., 2007*; *Mosimann et al., 2009*). The stabilization of Arm is therefore a critical step where Wg regulation can take place. An activated Wg signaling also can relocate Axin from intracellular vesicles to the plasma membrane, thereby inactivating the destruction complex and stabilizing Arm (*Cliffe et al., 2003*). Relocation of Axin through intracellular trafficking strengthens the fact that Wg signaling is equally regulated through a non-canonical mechanism. Intracellular trafficking affects the efficiency of Wg signaling significantly. For instance, Wg accumulates on the cell surface of Dynamin mutant clones (*Strigini and Cohen, 2000*). A Rab5 knockdown further down-regulates Wg signaling suggesting that an internalization and

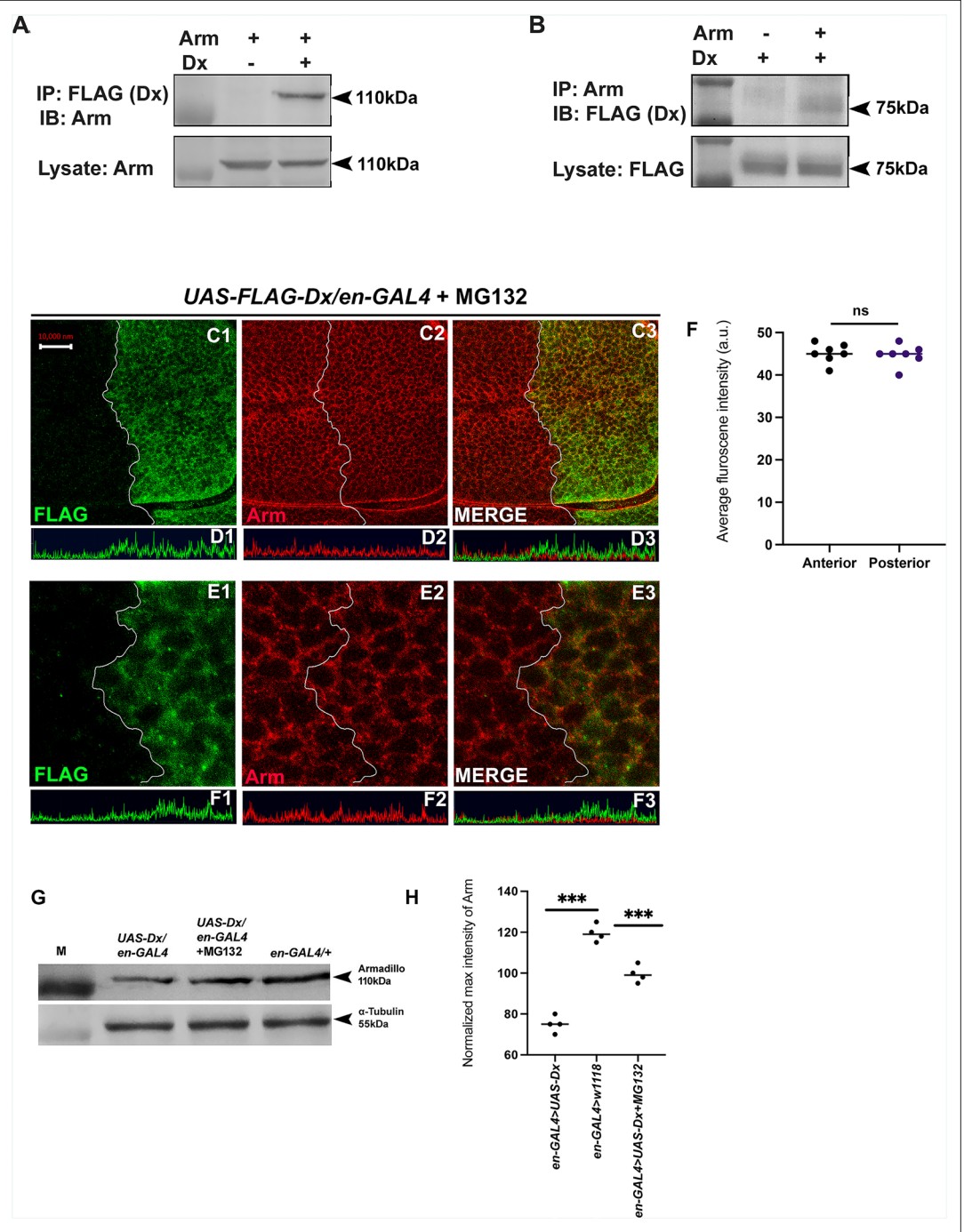

**Figure 7.** Dx degrades Arm by a proteasome-mediated mechanism. (**A, B**) Co-immunoprecipitation of FLAG-Dx and Arm. Co-immunoprecipitation was carried out with lysate over-expressing Arm and FLAG-Dx using *GMAR-GAL4*. + symbol indicates the presence of lysate and the − symbol shows the absence of lysate. FLAG-Dx immunoprecipitated Arm was detected by anti-Arm antibody (**A**). Arm immunoprecipitated FLAG-Dx was detected by FLAG antibody. (**C1–C3**) Dx over-expressed discs treated with MG132 for 3 hr showed a rescue in Arm expression and a more intense Arm staining in the posterior compartment was found after proteasome inhibitor treatment. (**D1–D3**) Confocal Z stack Intensity profiling shows a comparable Arm level in the posterior compartment of the wing imaginal disc. (**E1–E3**) A higher magnification picture shows no significant change in the expression of Arm in the posterior compartment of the disc suggesting the rescue in Arm degradation. (**F**) Graph shows the average Arm fluorescence intensity (a.u., arbitrary units) (*n* = 7). *ns*; unpaired *t*-test. (**G**) Western blot analysis confirms the immunocytochemical studies where proteasome inhibitor MG132 increases Arm protein

*Figure 7 continued on next page*

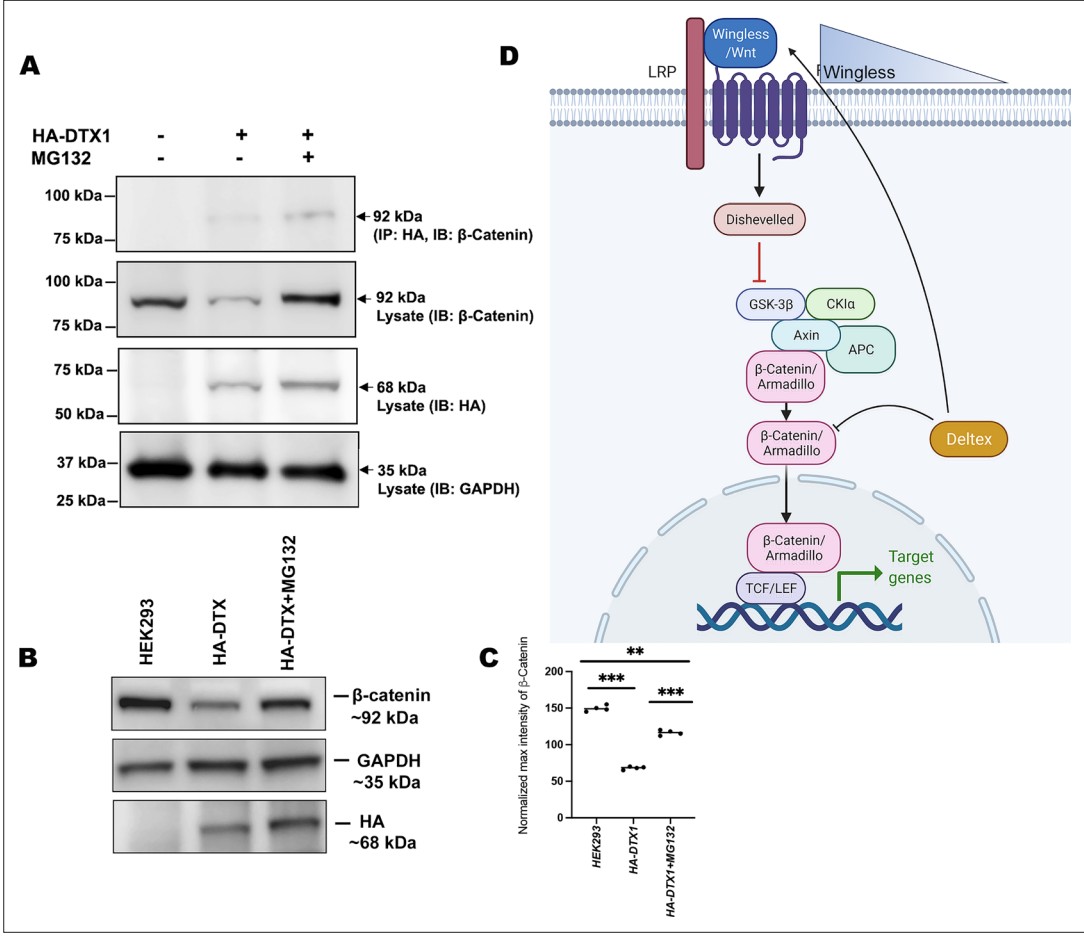

**Figure 8.** DTX1 physically interacts with β-catenin and facilitates its degradation. (**A**) HA-tagged human DTX1 transfected in HEK-293 cell line was pulled down with an anti-HA antibody. Immunoblot was done with β-catenin. A prominent band at 92 kDa corresponding to β-catenin post-MG132 treatment was observed. GAPDH was used as an internal control. (**B**) A rescue in β-catenin level post-MG132 treatment was observed. GAPDH serves as an internal control. (**C**) Graph C represents the intensity profiling of the Western blot in C. **p < 0.01, ***p < 0.001; one-way analysis of variance (ANOVA)/Turkey's multiple comparisons test. Images in A–F are representatives of three independent experiments. (**D**) The cartoon explains the two plausible mechanisms of Wg regulation through Dx. Dx facilitates Wg gradient formation on the one hand and on the other it targets Arm for its degradation thereby regulating the signaling output.

The online version of this article includes the following source data for figure 8:

**Source data 1.** Original file for the co-immunoprecipitation analysis in *Figure 8A* (anti-β catenin, anti-HA, and anti-GAPDH).

**Source data 2.** Original scans of the relevant co-immunoprecipitation analysis (anti-β catenin, anti-HA, and anti-GAPDH) with highlighted bands and sample labels for *Figure 8A*.

**Source data 3.** Original file for the Western blot analysis in *Figure 8B* (anti-β catenin, anti-HA, and anti-GAPDH).

**Source data 4.** Original scans of the relevant Western blot analysis (anti-β catenin, anti-HA, and anti-GAPDH) with highlighted bands and sample labels for *Figure 8B*.

---

endosomal transport of Wg is critical for its regulation (*Seto and Bellen, 2006*). Wg internalization is proposed to occur via two spatially distinct routes: one on the apical, and one on the basal, side of the disc. It is therefore thought that the subcellular localization of Wg along the apical–basal axis of receiving cells could play a crucial role in molding the Wg gradient (*Marois et al., 2006*).

*Figure 7 continued*

levels in Dx over-expressed tissue samples. (**H**) Graph G represents the intensity profiling of the Western blot. ***p < 0.001; unpaired *t*-test. Images in A–F are representatives of three independent experiments (*n* = 6). Scale bar: A1–A4: 10 μm. C1–C4: 2 μm.

The online version of this article includes the following source data for figure 7:

**Source data 1.** Original file for the co-immunoprecipitation analysis of *Figure 7A* (anti-Armadillo).

**Source data 2.** Original scans of the relevant co-immunoprecipitation analysis (anti-Armadillo) with highlighted bands and sample labels for *Figure 7A*.

**Source data 3.** Original file for the co-immunoprecipitation analysis of *Figure 7B* (anti-FLAG).

**Source data 4.** Original scans of the relevant co-immunoprecipitation analysis anti-FLAG with highlighted bands and sample labels for *Figure 7B*.

**Source data 5.** Original file for the Western blot analysis in *Figure 7G* (anti-Armadillo and anti-tubulin).

**Source data 6.** Original scans of the relevant Western blot analysis (anti-Armadillo and anti-tubulin) with highlighted bands and sample labels for *Figure 7G*.

---

Previously, Godzilla, a member of the RNF family of membrane-anchored E3 ligases is shown to be required for subsequent trafficking of Wg from early apical endosomes to the basolateral surface (*Yamazaki et al., 2016*). Here, we show that Dx plays an important role in the intracellular trafficking of Wg and consequently facilitates the spreading of the Wg gradient. Dx regulates Wg signaling activity both at the level of spreading of Wg ligand and directly at the level of Arm stabilization. We hypothesize that since Dx has an E3 ubiquitin ligase activity, it can directly degrade Arm through ubiquitination followed by proteasomal degradation and thus diminish Wg signaling. Dx is a cytoplasmic protein and is best known for its regulation of Notch signaling (*Hori et al., 2004*; *Matsuno et al., 1995*). Dx regulates Notch signaling in a non-canonical manner and in an over-expressed condition Dx depletes Notch from the cell surface and accumulates it in the endocytic vesicles (*Hori et al., 2004*). Involvement of Dx in signaling cascade other than Notch has been reported from our earlier studies, where we show that Dx interacts with the TNF (Tumor Necrosis Factor) ligand Eiger, and regulates JNK signaling by facilitating the re-localization of Eiger from the cell membrane to the cytoplasm (*Dutta et al., 2018*). Moreover, we have also reported that Dx synergizes with TRAF6, the adaptor molecule in the JNK cascade, and activates JNK signaling at a level downstream of the ligand–receptor interaction (*Sharma et al., 2021a*). A novel function of Dx in Toll pathway activation has also been reported lately, where we have reported a JNK-independent Toll pathway activation through Dx (*Sharma et al., 2021b*).

In this study, we present evidence for the first time that *dx* genetically interacts with *wg* and other components of the Wg signaling cascade in particular the transcription factor, *arm*. Our loss-of-function and gain-of-function studies show that Dx is involved in the regulation of Wg signaling. We show that over-expression of Dx plays an important role in promoting the spreading of the morphogen Wg. This leads to an erosion of the Wg gradient which consecutively results in a loss of the short-range targets, Sens and Cut. We also observed that the loss of Wg downstream targets upon Dx over-expression is independent of Notch. On the contrary, reducing the dosage of Dx narrows the gradient of Wg expression, supporting further the direct role of Dx in Wg spreading. Moreover, on knocking down Rab5, the small GTPase required for early-endosome formation, along with Dx over-expression, a reduction in wing size was observed with a reduction in the Wg expression. A similar reduction in Wg expression was observed when Rab7 was over-expressed along with Dx. Rab7 facilitates the late endosome formation, and along with Dx it might facilitate Wg trafficking and in turn, maintains its proper gradient.

It has been shown previously that Arm associates with Notch near the adherens junctions and that it is rapidly endocytosed promoting the traffic of an activated form of Armadillo into endosomal compartments, where it may be degraded (*Sanders et al., 2009*, *Acar et al., 2021*). Notch signaling regulates the activity and the amount of active/oncogenic form of Arm/β-catenin (*Hayward et al., 2005*). More specifically, in colon cancer cells Notch physically interacts with unphosphorylated β-catenin and negatively regulates the translational accumulation of active β-catenin (*Kwon et al., 2011*). These studies contemplate the diverse range of mechanisms through which Arm and consequently, Wg signaling may be intricately controlled. Here, we propose a model of Wg regulation through Dx

via down-regulation of Arm. Dx down-regulates Arm when it was over-expressed in the wing and eye tissue. Since Dx has an E3 ubiquitin ligase activity and therefore this down-regulation of Arm can be attributed to being a ubiquitination-dependent phenomenon. Indeed, our results support this hypothesis since post-MG132 treatment rescues the loss of Arm when Dx is over-expressed in the background. In addition, we have also shown that human Deltex (DTX1) also displays a conserved function of proteasomal degradation of β-catenin.

The importance of Wnt/Wg signaling is acknowledged by its implicit role in myriads of biological phenomena throughout development. Wnt/Wg signaling has been implicated in many physiological processes, including development, tissue homeostasis, and tissue regeneration (*Wodarz and Nusse, 1998*). Thus, it is no surprise that mutations in the Wnt pathway components are associated with many hereditary disorders and cancers such as colorectal cancer, Wilms tumor, Familiar Adenomatous Polyposis, and leukemia (*Kinzler et al., 1991*; *Nishisho et al., 1991*; *Polakis, 2007*; *Zhan et al., 2017*). Thus, understanding the mechanism of Wnt/Wg signal regulation is a potent area of investigation. Increased expression of DTX and its involvement in cell migration and invasiveness in different cancers such as follicular lymphomas, osteosarcoma, glioblastoma, and melanoma have also been reported earlier (*Gupta-Rossi et al., 2004*; *Huber et al., 2013*; *Zhang et al., 2010.*, *Bachmann et al., 2014.*, *Thang et al., 2015*). Here, we have uncovered an important function of the cytoplasmic protein Dx in the gradient formation of the morphogen Wg. These results strengthen the role of Dx in the trafficking of morphogens, thereby allowing the proper activation of the short- and long-range target genes. In addition, the E3 ubiquitin ligase activity of Dx further opens a new avenue of Wg signaling regulation by maintaining the critical concentration of the effector molecule Arm. A clear understanding of Dx-mediated regulation of Wg signaling will be very useful for future research in the field of therapeutic intervention of diseases involved in aberrant Wg signaling.

## Materials and methods
### *Drosophila* genetics

All fly stocks were maintained on standard cornmeal/yeast/molasses/agar medium at 25°C as per standard procedures. *UAS-NICD*, *UAS-FLAG-Dx*, *dx*, and *dx*[152] were kindly provided by Prof. Spyros Artavanis Tsakonas. *wg*[CX3] (BL-2977), *wg*[CX4] (BL-6908), *fz*[1] (BL-1676), *fz*[MB07478] (BL-25554), *Sens*[E58] (BL-5312), *UAS-arm*[S10] (BL-4782), *UAS-Rab5DN* (BL-9771), *UAS-Rab7-GFP* (BL-42706), *UAS-Dx-RNAi* (BL-35677), *NRE-GFP* (BL-30728), *UAS-p35*, *vg-lacZ*, *en-GAL4*, *dpp-GAL4*, *C96-GAL4*, *ap-GAL4*, *pnr-GAL4*, *dll-GAL4*, and *GMR-GAL4* stocks were obtained from Bloomington Stock Center.

All crosses were performed at 25°C unless otherwise mentioned. To induce the expression of the gene in the specific domain GAL4–UAS binary system was used (*Brand and Perrimon, 1993*). Combination stocks were made with the help of appropriate genetic crosses.

## Immunocytochemistry and microscopy

*Drosophila* third instar larval wing discs were dissected out in ice-cold 1× PBS (Phosphate-buffered saline), and tissues were fixed in a 1:1 mixture of 3% paraformaldehyde in PBS and at room temperature for 1 min, followed by a second fixation in 3% paraformaldehyde and 5% DMSO (dimethysulfoxide) for 20 min. Immunostaining was done as described previously (*Sharma et al., 2021a*). For extra-cellular Wg staining permeabilization of tissue was omitted. Third instar larval wing discs were incubated in ice with thrice the antibody concentration before fixation. Conventional staining protocol was followed thereafter. To inhibit proteasomal degradation, imaginal discs were incubated with 10 mM of MG132 (Sigma M7449) dissolved in Schneider Culture Media for 3 hr before proceeding with immunostaining.

The following primary antibodies were used in this study Mouse anti-Cut (1:00, DSHB-Developmental Studies Hybridoma Bank), Mouse anti-Wg (1:100, DSHB), Mouse anti-Arm (1:50, DSHB), Rabbit anti-Flag (1:00, Sigma), Guinea pig anti-Sens (1:10,000, kindly gifted by Prof. Hugo Bellen), Rabbit anti-Dpn (1:200, a generous gift from Prof. Yuh Nung Jan), guinea pig anti-Rab5 (1:1000; a generous gift from Prof. Akira Nakamura), Alexa488-, Alexa555-, or Alexa405-conjugated secondary antibodies (1:200, Molecular Probes) were used to detect the primary antibodies. Imaging was performed using a Carl Zeiss LSM 780 laser scanning confocal microscope and images were processed with Adobe Photoshop7.

## Plasmid and construct generation

DNA fragments coding for full-length human Deltex1 (1863bps) was inserted into the pcDNA3.1-HA (Addgene #128034) mammalian expression vector with an N-terminal HA tag. Forward primer 5′ CGCAGGGTACCATGTCACGGCCAGGCCACGGTG 3′ and reverse primer 5′ GCGAGTCTAGACTATC AAGCCTTGCCTGCAGCCTC 3′ were used for PCR (Polymerase chain reaction) amplification of full-length Deltex1 cDNA and inserted into pcDNA3.1-HA vector utilizing KpnI and XbaI restriction sites.

## Cell culture and transfection

HEK-293 (ATCC CRL-1573) was maintained in DMEM (Dulbecco's Modified Eagle Medium) supplemented with 10% fetal bovine serum and penicillin/streptomycin (Invitrogen). HEK293 (CRL-1573) cells were obtained from ATCC and tested for mycoplasma contamination. Detailed information about the cell line can be obtained from the ATCC site (here). Cells were cultured according to the manufacturer's instructions (ATCC). Plasmids transfection was done by Lipofectamine 2000 (Thermo Fisher Scientific) as described by the manufacturer.

## Co-immunoprecipitation and Western blotting

FLAG-tagged Dx was over-expressed in the wing discs under the control of *en-GAL4* driver. Wing discs were dissected and homogenized in 1× RIPA (Radioimmunoprecipitation assay) buffer (Cell Signaling Technology) containing PMSF (Phenylmethylsulfonyl fluoride) 1 mM (Sigma), and 1× Protease Inhibitor (Roche). For control, *en-GAL4/+* protein samples were used. To inhibit proteasomal degradation, discs were incubated with 10 mM of MG132 (Sigma M7449) dissolved in Schneider Culture Media for 3 hr before proceeding with sample preparation. Protein concentration was quantified by Nanodrop. 1 μg of protein was subjected to Western blotting using Immun-blot PVDF (Polyvinylidene fluoride) membrane (Bio-Rad). Mouse anti-Arm (1:1000), Mouse anti-β-tubulin (1:2000, DSHB), and anti-mouse IgG–AP conjugate (1:2000, Molecular probes) were used as primary and secondary antibodies to probe the blot. Color was developed, after washing the membrane three times in TBST (Tris-Buffered Saline), using Sigma FAST BCIP/NBT (Sigma).

For immunoprecipitation of Dx and Arm, protein lysates were prepared in 1× RIPA buffer (Millipore, #20-188) from *Drosophila* head tissue expressing *UAS-FLAG-Dx*, *UAS-arm$^{s10}$*, or both *UAS-arm$^{s10}$* and *UAS- FLAG-Dx* using a *GMR-GAL4* driver. Crude lysates containing 2 mg of total protein were mixed with 5 μl of rabbit anti-FLAG antibody or mouse anti-arm antibody and 30 μl of protein A/G beads before being incubated overnight with end-over-end rotation at 4°C. *UAS-arm$^{s10}$* or *UAS-FLAG-Dx* lysates were used as control samples. Beads were collected after washing thrice with 1× RIPA buffer and separated on a 12% denaturing SDS-PAGE (Sodium Dodecyl Sulphate-Polyacrylamide Gel) followed by transfer onto Immuno-blot PVDF membranes (Bio-Rad). Mouse anti-arm antibody at 1:1500 dilution or rabbit anti-FLAG antibody at 1:1000 dilution and goat anti-rabbit IgG–AP conjugate at 1:2000 or goat anti-mouse IgG–AP conjugate at 1:200 dilution (Molecular Probes) was used as primary and secondary antibodies to probe the blot. Colorimetric detection was performed using Sigma FAST BCIP/NBT.

For visualizing conservation across the phyla, human *Deltex1* was expressed in HEK- 293 cells by transient transfection as mentioned (see ). Proteasomal degradation was inhibited by subjecting the transfected cells to 50 μM MG132 for 4 hr. Protein samples were quantified for Western blotting. Immunoprecipitation was carried out as described by abcam immunoprecipitation kit (#ab206996). After washing the samples with IP buffer, samples were denatured and run in 4–20% Mini-PROTEAN TGX precast protein gels (Bio-Rad #4561098) followed by an overnight transfer onto a PVDF membrane. After blocking the membrane with 2.5% BSA (Bovine Serum Albumin) in TBST for 1 hr, the blots were charged with primary antibody followed by an HRP-conjugated secondary antibody. Signal was developed using Chemiluminescence (Thermo Fisher Scientific). Primary antibodies used for Immunoprecipitation and Immunoblotting were mouse monoclonal anti-HA (1:1000, Sigma), rabbit anti-β-catenin (1:1000, Cell Signaling), and Rabbit anti-GAPDH (1:1000, Cell Signaling). Images were obtained with Chemidoc XRS+ (Bio-Rad).

## Imaging of adult tissues and eye imprints

For adult eye imaging, the anesthetized flies of the desired genotype were kept on a bridge slide to maintain a proper angle. Documentation was done with the Leica MZ105 system. Eye imprints using

nail polish were made for analyzing ommatidial defects and were examined under differential interference contrast optics in a Nikon Eclipse Ni microscope (*Arya and Lakhotia, 2006*). For wing and leg wholemount preparation, adult tissues from the desired genotype were removed with the help of a needle and scalpel and cleaned with isopropanol before mounting under coverslips in the mounting media. The slides were observed under a bright-field microscope and imaging was done at proper magnification.

## Statistical analysis

For measuring wing and tarsal segment size, the area of the wing and the tarsal region were measured in arbitrary units using ImageJ software. The size of the wild-type wing and leg was considered as 100% and the size of wing and legs from other genotypes were measured in percent ratio to the wild-type wings and legs. For Arm staining, intensity per unit area of the discs was measured along the anterior–posterior axis using ImageJ software, and the graphs were produced using GraphPad Prism 5 software. Each dataset was repeated at least three times. $T$-test and one-way analysis of variance with Tukey's multiple comparison post-test were employed to determine the significance of the level of difference among the different genotypes. $p$-value $<0.05$ was accepted as statistically significant.

## Acknowledgements

The authors extend sincere thanks to Prof. Spyros Artavanis-Tsakonas (Harvard Medical School), Prof. Florencio Serras (University of Barcelona), Prof. Hugo Bellen (Baylor College of Medicine), Prof. Yuh Nung Jan (University of California, San Francisco), Prof. Akira Nakamura (Institute of Molecular Embryology and Genetics, Kumamoto, Japan), and the Bloomington Stock Centre for fly stocks and antibody. Some of the antibodies used in the work were obtained from Developmental Studies Hybridoma Bank, University of Iowa. We also acknowledge the confocal facility of DBT-BHU-ISLS, Banaras Hindu University. Artwork is created with BioRender (BioRender.com).

## Additional information

### Funding

| Funder | Grant reference number | Author |
|---|---|---|
| Institute of Eminence Scheme, Banaras Hindu University, India and Department of Science and Technology, Ministry of Science and Technology, India. | CRG/2021/006975 | Ashim Mukherjee |

The funders had no role in study design, data collection, and interpretation, or the decision to submit the work for publication.

### Author contributions

Vartika Sharma, Conceptualization, Data curation, Formal analysis, Validation, Visualization, Methodology, Writing - original draft, Writing – review and editing; Nalani Sachan, Data curation, Methodology, Writing – review and editing; Bappi Sarkar, Data curation, Methodology; Mousumi Mutsuddi, Supervision, Project administration, Writing – review and editing; Ashim Mukherjee, Conceptualization, Supervision, Funding acquisition, Investigation, Project administration, Writing – review and editing

### Author ORCIDs

Vartika Sharma http://orcid.org/0000-0001-6642-4277
Ashim Mukherjee http://orcid.org/0000-0002-3523-7089

Reviewer #1 (Public Review): https://doi.org/10.7554/eLife.88466.3.sa1
Reviewer #2 (Public Review): https://doi.org/10.7554/eLife.88466.3.sa2

Author response https://doi.org/10.7554/eLife.88466.3.sa3

## Additional files

### Supplementary files
• MDAR checklist

### Data availability
We do not have any datasets associated with this article. Source data files have been provided for Figures 7 and 8.

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
