## [Editor Report · eLife assessment]

This is a **useful** study of the connection between the ubiquitin ligase protein deltex and the wingless signaling pathway. Two different links are inferred from genetic interactions in vivo between loss-of-function mutations and overexpression. While the genetic data are **solid**, the precise mechanism underlying either effect remains to be established.

---

## [Referee Report · Reviewer #1 (Public Review)]

This study presents a genetic and molecular analysis of the role of the cytoplasmic ub ligase Deltex (Dx) in regulating the *Drosophila* Wingless (Wg) pathway in the larval wing disc. The study exploits the strength of the fly system to uncover a series of genetic interactions between dx and wg and fz allele that support a role for Dx upstream of the Wg pathway. These are paired with molecular evidence that dx lof alleles lower Wg protein in 'source' cells at the DV margin, and that Dx associates with Arm and lowers its levels in a manner that can be rescued by pharmacological inhibition of the proteasome. The genetic data are solid but subject to alternative explanations based on the authors' model that Dx both inhibits and activates the pathway. The molecular data are suggestive, but need follow up tests of how Dx affects Wg spread, and how Dx mediates poly-ub of Arm, and the degree to which Dx shares this role with the validated Arm E3 ligase Slmb. Overall, the story is very interesting but has mechanistic gaps that lead to speculative models that require more rigorous study to clarify mechanism. Dx sharing a role in Arm degradation with the Slmb/APC destruction would have important implications for the many Wg/Wnt regulated processes in development and disease.

---

## [Referee Report · Reviewer #2 (Public Review)]

The manuscript investigates the connections between the ubiquitin ligase protein deltex and the wingless pathway. Two different connections are proposed, one is function of deltex to modulate the gradient of wingless diffusion and hence modulate the spatial patter of wingless pathway targets, which regulate at different thresholds of wingless concentration. The second is a direct interaction between deltex and armadillo, a downstream component of the wingless pathway. Deltex is proposed to cause the degradation of armadillo resulting in suppression of wingless pathway activity. The results and conclusions of the manuscript are interesting and for the most part novel, although previously published work linking Notch and deltex to wingless signal regulation, and endocytosis to wingless gradient formation could be more extensively discussed. However neither of the two parts to the manuscript seem, in themselves sufficiently complete, and combining both parts together therefore seems to lack focus.

The main issue with the manuscript is that much of the conclusions are inferred from genetic interactions in vivo between loss of function mutants and overexpression. While providing useful in vivo physiological context, this type of approach struggles to be able to make definitive conclusions on whether an interaction is due to direct or indirect mechanism, as the authors themselves conclude at the end of section 2.3. The problem is confounded by the fact that there is already documented much cross talk between the Notch signaling pathway and wingless at the transcriptional level, and deltex is already a Notch modulator that can alter wingless mRNA expression (See Hori et al 2004). Deltex in addition to promoting a ligand-independent Notch signal can also induce expression of Notch ligand, allowing further non-autonomous Notch activation and subsequent cell autonomous cis-inhibition of the initial deltex-induced signal. The dynamics and outcomes of the Notch signal response to deltex in vivo is therefore already very complicated to interpret before even considering to unravel indirect (via Notch) and direct interactions with wingless, although the two possibilities are not mutually exclusive. Whilst the revised manuscript does not completely overcome these limitations, further data and quantification have improved the support for the conclusions and there is a wider discussion of the relevant literature. The conclusions are interesting and add significantly to our understanding of the intersections between Wingless, Notch and trafficking regulators in an in vivo context.

---

## [Author Response]

The following is the authors’ response to the original reviews.

**Reviewer #1 (Recommendations For The Authors):**
(1) The description of the wing phenotype that results from combinations of wingless and delex alleles at the bottom of page 4 (figure 1) is quite confusing. Are the trans-hets suppressed to wt or enhanced? The images in the Fig look enhanced.

We thank the reviewer for this thoughtful observation regarding the wing phenotype description in combination with wg and dx alleles. We understand the confusion and appreciate the opportunity to clarify.

In response to the concern raised, the trans-heterozygous indeed enhanced rather than suppressed to wild type. We acknowledge that the description would have been clearer. We have revised the relevant section to explicitly state that trans-heterozygous exhibit an enhanced wing phenotype in the updated version of the manuscript.

(2) Use of Cut as a Wg readout in Fig1 is problematic since it is also a Notch target. Perhaps a more direct measure of Arm activity would be a better choice here, e.g., naked-lacZ.

We appreciate the reviewer’s insightful comment regarding the use of Cut as a Wg readout. The point about being Cut as a Notch target raises a valid concern. To address this issue and provide a more direct measurement of Arm activity, we agree that incorporating a specific Arm readout, such as naked lacZ, would be a more suitable choice.

We will incorporate this valuable feedback into our future research endeavors to augment the comprehensiveness of our study.

(3) The dx allele effects on Sens and Vg in Fig 2C appear greater at two points along the DV margin (arrows). Do these match the expression pattern of dx mRNA?

We thank the reviewer for this thoughtful observation. We understand that the effect of the dx LOF allele on Sens and Vg seems more pronounced at two specific points along the D/V margin. As far as our understanding Dx shows a homogeneous expression pattern throughout the Wg disc which has been reported earlier (Busseau et al., 1994., Mukherjee et al., 2005).

(4) It really looks to my eye that dx loss lowers Wg expression in source cells in Fig 2. To confirm the model that Dx controls the spread of Wg protein, it would be ideal to rule out txnal effects with a wg-lacZ reporter.

We appreciate the reviewer for raising this important point. In the revised version of the manuscript, we have introduced Wg-lacZ staining for both Wg-lacZ/+ and dx152/Y; Wg-lacZ/+ combination in Figure 2. This additional information eliminates the possibility of Deltex influencing Wg transcriptional regulation in source cells, thus reinforcing our hypothesis that the reduction of Deltex leads to a decline in Wg protein levels in the source cells, given Dx essential role in wingless gradient formation.

(5) The drop in DV Wg and expansion of Vg domain in dx mutants seem paradoxical but could be explained by accelerated Wg spread and uptake. This could be tested by depleting the dally-like glypican that promotes long-range Wg diffusion in dx mutants, and seeing if this restores Wg levels at the DV margin.

This is indeed a very thoughtful comment and we thank the reviewer for this insightful suggestion for further exploration. We believe that depleting dally-like glypican in dx mutants could possibly restore Wg levels at the DV margin.

We recognize the importance of this experiment in providing a more comprehensive understanding of the underlying mechanisms, and we will give major emphasis on incorporating this suggestion in our future research.

(6) The authors describe the effect of Dx over-expression as "reducing" the Wg gradient when they actually mean "flattening". Please be careful with this word choice as they mean different things.

We thank the reviewer for the insightful feedback. The suggested modifications have been incorporated into the revised version of the manuscript.

(7) The combined effects of Rab5dn and Dx o/e on Wg protein loc/levels are interesting but need to be followed up by testing whether the endogenous Dx/Rab5 show genetic interactions in control of Wg protein levels/localization.

We acknowledge the reviewer's comment and in addressing it, we wish to highlight that the over-expression of Dx with endogenous Rab5 or Rab7 does not affect Wg protein levels or localization. We have mentioned the supporting data for this control in Figure 5(G, H).

(8) The ability of MG132 to restore Arm levels in en-Dx discs is very promising. However, MG132 will also block Arm degradation by the Slmb-APC destruction complex, so this result could be non-specific. Tests of whether Dx drives poly-ub of Arm, and how much Dx is redundant to Slmb in this role, would be needed to solidify the authors' conclusion.

We thank the reviewer for this insightful comment. We understand that the concern about MG132 blocking Arm degradation by Slmb-APC destruction complex adds an important layer of complexity to the interpretation of the results. We agree with the reviewer's comment that conducting these experiments will indeed offer valuable insight into the specificity of MG132 effects and further strengthen our conclusion.

We are interested to see how future experiments addressing the points raised by the reviewer will shape our understanding of the intricate mechanisms involved in Wg signaling and Arm/β-catenin degradation. Once again, we thank the reviewer for the thoughtful engagement with the research, and the comments will undoubtedly stimulate further investigation and discussion in this area.

**Reviewer #2 (Recommendations For The Authors):**
The work really needs more experiments to further provide a mechanistic understanding and distinguish between direct and indirect action (via Notch signaling) on Wingless, but instead switches in the second half to a second interaction with β-catenin, leaving the conclusions of the first part hanging. More mechanistic information on the cell biology of how Deltex might affect wingless endocytic trafficking directly would be beneficial, for example involving some cell culture experiments where the action of deltex on Notch and wingless could be more clearly separated and a more detailed study of the consequences on wingless trafficking could be explored.Wingless is secreted into an extracellular compartment and so won't be accessible for a direct interaction with cytoplasmic deltex. Therefore are the authors proposing Deltex interacts with a membrane-bound wingless receptor such as frizzled in order to mediate its effects? These avenues could be explored further experimentally to derive a more mechanistic conclusion.The colocalisation images are not high resolution and colocalisation is not quantified, and no differences ( +/- Deltex) in wingless subcellular localisation, which would aid mechanistic interpretation, are shown.

We thank the reviewer for the insightful feedback on our work. We appreciate the suggestion for more experiments to provide a mechanistic understanding and to distinguish between direct and indirect actions of Notch on Wingless signaling. We acknowledge the importance of clarifying these aspects and agree that further experiments could help separate the effects of Deltex on Notch and Wingless signaling, allowing for a more detailed examination of their respective trafficking and ubiquitination mechanisms.

We will consider your valuable input in our future research efforts to enhance the comprehensiveness of our study.

Other specific pointsFigure 2: Narrowing and broadening of different marker gene expression patterns in dx mutants needs to be quantified so that variation is taken into account and the numbers of wings imaged should be clearly stated.

We greatly appreciate this valuable suggestion from the reviewer. As a response, we have incorporated quantification data to address the observed variations. We have also provided information regarding the number of wing discs that were imaged for the purpose of quantification.

Figure 3: The number of discs imaged in total should be mentioned

We express our appreciation to the reviewer for the input. We have taken their comment into account and have subsequently included details regarding the number of discs imaged in the figure legend section of the manuscript.

Figure 6: There is no description of (E5-E6) in the figure legend. F1 to F5 eye size phenotypes require quantification.

We are grateful to the reviewers for bringing this to our attention. In response, we have included a description of E5-E6 in the figure legend. Also, as per the reviewer’s suggestions, we have incorporated the quantification data of the eye size phenotype.

DiscussionLinks between Notch and wingless pathway should be more comprehensively discussed, including previous work that has previously linked Notch/Deltex to β-catenin degradation e.g.Acar et al. .Sci Rep 2021 Apr 27;11(1):9096. doi: 10.1038/s41598-021-88618-5Hayward et al. Development 2005 Apr;132(8):1819-30. doi: 10.1242/dev.01724;Kwon et al Nat Cell Biol 2011 Aug 14;13(10):1244-51. doi: 10.1038/ncb2313.Sanders et al. PLoS Biol 2009 Aug;7(8):e1000169. doi:10.1371/journal.pbio.1000169. Epub 2009 Aug 11.The links between endocytic trafficking and wingless gradient formation could also be further discussed eg.Marois et al. Development 2006 Jan;133(2):307-17.doi: 10.1242/dev.02197. Epub 2005 Dec 14Yamazaki et al Nat Cell Biol 2016 Apr;18(4):451-7. doi: 10.1038/ncb3325. Epub 2016 Mar 14.

We appreciate the reviewer's valuable suggestions and we have now included these references in the discussion section of the revised manuscript.